# Microbiota Profile of the Nasal Cavity According to Lifestyles in Healthy Adults in Santiago, Chile

**DOI:** 10.3390/microorganisms11071635

**Published:** 2023-06-22

**Authors:** Daniela Toro-Ascuy, Juan P. Cárdenas, Francisco Zorondo-Rodríguez, Damariz González, Evelyn Silva-Moreno, Carlos Puebla, Alexia Nunez-Parra, Sebastián Reyes-Cerpa, Loreto F. Fuenzalida

**Affiliations:** 1Facultad de Ciencias de la Salud, Instituto de Ciencias Biomédicas, Universidad Autónoma de Chile, Santiago 8910060, Chile; daniela.toro@uautonoma.cl (D.T.-A.); evelyn.silva@umayor.cl (E.S.-M.); carlos.puebla@uoh.cl (C.P.); alexia.nunez@uchile.cl (A.N.-P.); 2Centro de Genómica y Bioinformática, Facultad de Ciencias, Ingeniería y Tecnología, Universidad Mayor, Santiago 8580745, Chile; juan.cardenas@umayor.cl (J.P.C.); damigonzalez38@gmail.com (D.G.); 3Escuela de Biotecnología, Facultad de Ciencias, Ingeniería y Tecnología, Universidad Mayor, Santiago 8580745, Chile; 4Departamento de Gestión Agraria, Facultad Tecnológica, Universidad de Santiago de Chile, Santiago 8910060, Chile; francisco.zorondo@usach.cl

**Keywords:** microbiota profile, upper respiratory tract, nose, healthy adults, sex, nutritional status

## Abstract

Background: The respiratory microbiome is dynamic, varying between anatomical niches, and it is affected by various host and environmental factors, one of which is lifestyle. Few studies have characterized the upper respiratory tract microbiome profile according to lifestyle. We explored the association between lifestyles and microbiota profiles in the upper respiratory tract of healthy adults. Methods: We analyzed nasal samples from 110 healthy adults who were living in Santiago, Chile, using 16S ribosomal RNA gene-sequencing methods. Volunteers completed a structured questionnaire about lifestyle. Results: The composition and abundance of taxonomic groups varied across lifestyle attributes. Additionally, multivariate models suggested that alpha diversity varied in the function of physical activity, nutritional status, smoking, and the interaction between nutritional status and smoking, although the significant impact of those variables varied between women and men. Although physical activity and nutritional status were significantly associated with all indexes of alpha diversity among women, the diversity of microbiota among men was associated with smoking and the interaction between nutritional status and smoking. Conclusions: The alpha diversity of nasal microbiota is associated with lifestyle attributes, but these associations depend on sex and nutritional status. Our results suggest that future studies of the airway microbiome may provide a better resolution if data are stratified for differences in sex and nutritional status.

## 1. Introduction

The human microbiota is composed of thousands of different microorganism genes and different bacterial communities that are known to be involved in a diverse array of physiological functions, such as immune system development [1,2,3], nutrition [4,5], and resistance to colonization by pathogens [6,7], among others.

The microbiota can be determined by people’s lifestyles. Some factors studied are diet and stress, physical activity, and smoking habits both in the gut and respiratory microbiota [8,9]. In a recent study, the nasopharyngeal profile of young children with overnutrition was characterized by an over-representation of pathogenic bacteria and proinflammatory cytokines [10].

Although intestinal microbiota has been extensively investigated [11,12,13,14,15], fewer studies have characterized the microbiome profile of the upper respiratory tract, particularly the nostrils. Understanding the microbiota and how it may change is relevant due to its role in physiological functions, especially in healthy people. Far from the historical concept of a sterile environment of the lungs, the lung microbiota significantly contributes to airway tolerance and immune responses to respiratory infection [16,17]. The upper respiratory tract is now recognized as a reservoir of pathogens, and some studies have suggested that it may play a role in the development of allergies and asthma by modifying airway mucosal inflammation and stimulating the formation of the upper respiratory tract immune regulation [1,18]. However, the microbiota could also prevent the entry of pathogens [19], implying that different conditions could determine how the upper respiratory tract microbiota interacts with its host. Specific sites in the respiratory tract contain specialized bacterial communities that are thought to have a significant role in human health, but also intervene in the anatomical development and maturation of the human respiratory tract, both in prenatal and postnatal life. Similarly, upper respiratory microbiota plays an important role in immune training, organogenesis, and the maintenance of immune tolerance, suggesting that adequate microbiota sensing is essential for immune maturation and consecutive respiratory health [20]. For example, the immunomodulatory species of *Prevotella* promote lung homeostasis. The pro-inflammatory environment in the lung enhances the growth of various Gram-negative bacteria, such as gammaproteobacteria, through nutrient enrichment processes [21].

With the current SARS-CoV-2 pandemic, the respiratory tract microbiota has become even more relevant. A review suggested associations between gut and respiratory tract microbiota, host antiviral immunity, and influences of dietary, nutritional, and lifestyle interventions in preventing the clinical course toward more severe SARS-CoV-2 disease [22]. Moreover, a sex gap exists in respiratory diseases such as cystic fibrosis and chronic obstructive pulmonary disease [23,24] where the pathophysiology of the sex difference has been poorly characterized to date. In recent years, studies have revealed the presence of a resident microbiome in the respiratory tract and its central role in respiratory disease, achieving a new way to explore and understand the observed sex gap in respiratory diseases.

Therefore, identifying the type of bacteria present and determining their relative abundance will provide critical information on aspects of the microbiota that correlate with human health and lifestyles. To achieve potentially effective treatments against respiratory pathogens, it will be necessary to understand the dynamics of the upper respiratory microbiota profile, its interaction with the host, and the environmental factors that modify it.

In this study, our main goal was to analyze the microbiota profile of the upper respiratory tract and estimate its association with the lifestyle attributes of healthy adults. We also compared the associations of microbiota profiles and lifestyle attributes between men and women, as well as between different nutritional statuses. As a case study, we used samples of healthy adults living in Santiago, Chile.

## 2. Materials and Methods

### 2.1. Study Design and Participants

A total of 110 urban volunteers who were living in Santiago, Chile, were recruited by the citizen science project carried out at the Universidad Autónoma de Chile in 2017. Participants were selected based on the following inclusion criteria: healthy adults (over 18 years of age) who did not have a respiratory infection at the time of sample collection and who had not taken an antibiotic in the previous two weeks were included. No exclusion criteria were considered. Volunteers were asked to (i) complete a structured questionnaire, and (ii) collect a nasal sample. The questionnaire gathered information on sociodemographic and lifestyle attributes, including physical activity (classified as sedentary, active, and vigorous lifestyle, according to the World Health Organization (WHO) [25], size and weight, smoking, medication use, alcohol consumption, sex, and age (Appendix A)). Nutritional status was determined by BMI (body mass index) stratified by the WHO, defined as: underweight (<18.5 kg/m^2^), normal weight (18.5–24.9 kg/m^2^), overweight (25.0–29.9 kg/m^2^), and obese (≥30.0 kg/m^2^) [26]. In total, 110 out of 120 samples were selected based on DNA quality.

### 2.2. DNA Extraction and 16S rDNA Gene Sequencing

DNA extraction, sequencing, and taxonomic assignments were performed at the uBiome facilities (San Francisco, CA, USA) in March 2018 according to previous instructions [27]. Amplicons analysis and taxonomical assignment were also performed as previously established [27,28]. After sequencing, forward and reversed 16S rRNA gene V4 sequence reads were demultiplexed, and reads were filtered using an average Q-score > 30 and merged after primer removal. The most abundant sequence per cluster was considered the real biological sequence and was assigned the count of all reads in the cluster. Only samples with at least 10,000 high-quality reads were used in the analysis [28]. Chimeras were removed using VSEARCH [29] and clustered using Swarm [30]. The resulting clusters were then compared to a curated version of the SILVA database, release 132.0 (https://www.arb-silva.de/documentation/release-132, accessed on 15 June 2023) [31], using 100% identity over 100% of the length. The relative abundance of each taxon was determined by dividing the count linked to that taxa by the total number of filtered reads [27], performed following the pipelines described by Almonacid et al. [27] and Bik et al. [28]. In both cases, the experimental procedure followed the Standard Operating Procedures in a Clinical Laboratory Improvement Amendments (CLIA) licensed and College of American Pathologists (CAP)-accredited laboratory (as mentioned by Vera-Wolf et al. [32]). Since the taxonomic assignment was made from comparisons “using 100% identity over 100% of the length”, this method was very conservative for the assignments and served to discard sequence contaminations.

### 2.3. Microbiota Data Analyses

The matrix comparing microbial taxonomic groups among samples (from superkingdom to genus levels) was used for PCA analysis using the prcomp instruction in R. The relationships between samples and variables were plotted using the factoextra package of R. Hierarchical clustering (average linkage) and dendrograms were elaborated using the hclust instruction and the ggdendro package (v. 0.1.23) from R v.4.2. Plots were elaborated using the ggplot2 package in R v.4.2.

Taxonomic relative abundance matrices were utilized to search for taxonomic markers via LefSe v.1.1.2 [33] using default options (involving a *p*-value > 0.05 and an LDA > 2 or LDA < −2). The classes considered in different comparisons were: sex (female/male), smoking (smoker/non-smoker), alcohol consumption (yes/no), or nutritional status (normal-low weight/overweight–obese).

### 2.4. Microbial Diversity Analysis

Alpha diversity was calculated for all taxonomic groups, using Shannon entropy H, the number equivalent of taxonomic groups (exp H), Simpson’s sum of squared probabilities R, and the inverse of Simpson R. We used the “entropyetc” package [34] in Stata 14.1 (Statacorp, 2016, College Station, TX, USA). The indexes were calculated for all participants in the study.

### 2.5. Statistical Analyses

Ordinary least-squares multivariate models were adjusted to estimate the associations between indexes of alpha diversity, as outcome variables, and participants’ attributes, as explanatory variables, such as physical activity, nutritional status, smoking, medication, consumption of alcohol, age, and sex. Different interactions between the explanatory variables were tested in the multivariate regressions. Since the interaction between nutritional status and smoking was significantly associated with the alpha diversity indexes, it was the only interaction included in the informed model. Hube and White’s estimator of variance was used to obtain robust standard errors. The models were adjusted for all 110 samples, differently for sex (63 women and 44 men) and for nutritional status (66 underweight–normal weight and 44 overweight–obese). The level of significance was set at 5% with a confidence interval of 95%. The analysis was performed in Stata 14.1 (StataCorp, 2016).

### 2.6. Ethical Regulations

This study was approved by the ethics committees of the Universidad Autónoma de Chile (CEC 52-22). An exception of informed consent was solicited and approved. The use of lifestyle and microbiological databases included in this study did not violate the dignity of the subjects that could have been involved, ensured the right to privacy and anonymity, and guaranteed the protection of the confidentiality of the data and the custody of the information and the use that was given to it.

## 3. Results

### 3.1. Participant’s Sociodemographic and Lifestyle Attributes

In total, 63 (or 57%) out of the 110 participants who joined the study were female, and the average age of the participants was 33.7 (std. dev. = 10.9, range 18–71) years old (Appendix A, Participant’s information). On average, the BMI of the participants was 24.4 (std. dev. = 3.5) kg/m^2^. According to nutritional status, 65 participants out of 110 were classified as normal weight (60%), whereas 36 (33%) and 8 (7%) participants were classified as overweight and obese, respectively. Only one person had a BMI (18.3) slightly lower than 18.5, and he was included in the group of normal weight for the statistical analyses. Regarding physical activity, 70 (64%) out of the 110 participants reported engaging in sedentary activity and 19 participants declared engaging in active activity. Twenty-one persons reported vigorous physical activity. Seventy-six (69%) participants reported consuming alcoholic drinks, whereas eighteen (16%) participants were smokers.

### 3.2. Microbiota Characteristics of the Nose with Overall Properties

The first approach to studying the deep amplicon sequencing-derived samples was the taxonomic assignment analysis, using the uBiome pipeline (see Section 2, Materials and Methods). The different phyla and families represented in the 110 samples are represented in Figure 1. The taxonomic assignment results revealed that most samples were dominated by members of the Proteobacteria, Firmicutes, and Actinobacteria phyla (Figure 1A). Other phyla represented in some samples included Fusobacteria, Bacteroides, and Acidobacteria. Thaumarcheota was the archaeal phylum primarily found among samples. Despite that general trend, some samples showed strong dominance of one particular group: some samples were mainly composed of one of Firmicutes, Actinobacteria, or Proteobacteria. According to the same analysis, the most important families detected among samples (Figure 1B) were Corynebacteriaceae, Staphylococcaceae, Moraxellaceae, and Carnobacteriaceae, and to a lesser extent, Leuconostocaceae, Comamonadaceae, Propionibacteriaceae, and Brucellaceae. In the case of taxonomic genera, only 17 of them were found in more than 10% of the samples (Table 1). Six of them (*Staphylococcus* and *Anaerobacillus* from Firmicutes, *Corynebacterium* and *Propionibacterium* from Actinobacteria, and *Delftia* and *Ochrobactrum* from Proteobacteria) were found in more than half of the samples. It was noticeable that some of the 17 genera had high standard deviations in comparison to their medians, suggesting a very high heterogeneity in their abundance.

### 3.3. Differences PCA between Lifestyle Attributes at Cluster Level

The information from all taxonomic predictions among the samples (except those at the species level) was used for principal component analysis (PCA) in order to reduce the dimensionality of the data and to visualize differences between the samples. This analysis revealed a “triangular”-like distribution of the samples based on their taxonomic composition; the first two principal components could explain 44.04% and 29.48% of the total variation, respectively (Appendix A). When k-means (k = 6) clustering was applied according to the results of a sample cladogram, the clustering patterns in the plot did not have an obvious pattern (Appendix A). In agreement with this clustering pattern, if samples distributed in the PCA profile were colored by different aspects of their metadata, such as sex (Appendix A), nutritional status (Appendix A), alcohol use (Appendix A), or physical activity (Appendix A), they did not reveal any evident clustering pattern either. This could also be a signal of the high heterogeneity of microbial composition among the samples.

In order to explore which variables were the most relevant in how samples are separated, the variables were interrogated using the “fviz_pca_biplot” instruction of the R *factoextra* package (Appendix A). These results suggest that there are three main directions influencing the shifting of the samples through the plane, influencing the previously mentioned “triangular”-like distribution: the first shifting effect, produced towards the negative side of PC2 and the negative side of PC1, was driven by Proteobacteria and several derived groups (mainly Betaproteobacteria, Burkholderiales, Comamonadaceae, *Delftia*, Alphaproteobacteria, and Rhizobiales) as well as *Anaerobacillus* and Bacillaceae from Firmicutes. The second one, produced towards the negative side of the PC2, was driven by Firmicutes and other related groups, such as Bacillales, Bacilli, *Dolosigranulum*, Carnobacteriaceae, Lactobacillales, *Staphylococcus*, and Staphylococcaceae. Finally, the third shift (towards the positive coordinates of PC1) was driven by Corynebacteriaceae and Actinobacteridae. This may reflect that the microbial composition of some taxonomic groups is more relevant for sample differentiation rather than the properties of the human subjects. This may reflect how internal properties in the study group could lead to the predominance of different characteristic groups.

### 3.4. LDA: Linear Discrimination Analysis and Principal Components Analysis at Multilevel

Previous results have suggested the presence of different taxonomic groups as “shifters” in the dimensional reduction in sample composition. These results also suggested that there are no clear clusters from the results of PCA. In order to establish specific taxonomic markers comparing different groups, compositional data were analyzed by the use of LefSe (Linear discriminant analysis effect Size). LefSe is a commonly used method for identifying differentially abundant features between two or more groups in microbiome datasets using linear discrimination analysis (LDA) tool LefSe [33]. Each analysis was conducted with default parameters since these values (alpha < 0.05 and LDA < 2) have already been evaluated as strong parameters to ensure very low false positive rates [33]. Additionally, no sub-classes were defined in each analysis to ensure that inter-class comparison considered each sample without the need to search for internal differences among sub-classes.

In this study, several dichotomic comparisons were performed from compositional data among different metadata criteria. The first comparison was performed between samples from both women and men (Appendix A). Interestingly, LDA showed taxonomic markers differentiating male and female samples: Lachnospiraceae and *Blautia* were found as markers in female samples, whereas a set of 20 taxonomic groups (including the *Prevotella*, *Actinobacillus*, *Gemella*, *Finegoldia*, *Streptococcus*, *Anaerococcus*, *Peptoniphilus*, and *Corynebacterium* genera) were found as sex-specific markers from male samples. Comparing samples with different nutritional states and grouping normal and low-weight subjects with all subjects with a higher BMI (overweight or obese), we could observe that the taxonomic groups *Haemophilus*, Bacteroidia, Bacteroidales, *Leuconostoc*, and Leuconostocaceae were markers associated with normal weight; conversely, Negativicutes, Selenomonadales, Veillonellaceae, *Rothia*, Lachnospiraceae, Pasteurellaceae, Pasteurellales, *Streptococcus*, Streptococcaceae, and *Moraxella* were found as markers for obese–overweight samples. Other comparisons were also made in the general population set (Appendix A): comparing samples from people with low-sedentary activity and people with active-vigorous activity identified that several groups associated with the Proteobacteria phylum (the gamma- and alpha-proteobacteria classes, and the *Ralstonia* and *Sphingomonas* genera, among other groups) were found as markers for sedentary activity, but no marker was found for active-vigorous activity. When comparing smokers and non-smokers, Acidobacteria and related groups (at different taxonomic levels) were found as markers for the smoking lifestyle. *Citrobacter* was found as the only marker group when we compared microbiota samples from alcohol consumers with samples from non-alcohol consumers.

The presence of several different markers by sex suggests that the possibility of the differential behavior of other metadata aspects (e.g., normal-low weight vs. overweight–obese) is influenced by the host’s sex. In order to evaluate this possibility, we separately performed LDA in men-only and women-only samples against other behavior statuses (Appendix A, respectively). Interestingly, we noticed that several markers observed in women-only and men-only samples were present in the overall comparison, as mentioned in the paragraph above. For example, when nutritional status (low-normal vs. overweight-obese) was compared, in male-only samples, five taxonomic groups (*Rothia*, Micrococcineae, *Moraxella*, Moraxellaceae, Pseudomonadales) were found as markers associated with the obese–overweight group, whereas in female-only samples, ten markers were found to be associated with the obese–overweight status (*Haemophilus*, Selenomonadales, Negativicutes, Veillonellaceae, Pasteurellales, Pasteurellaceae, *Blautia*, *Rothia*, Lachnospiraceae, and Corynebacterineae). Interestingly, *Rothia* (separately found in female and male samples) was also found in the general comparison. Moreover, *Haemophilus* (found as a marker of normal weight in the general comparison) was found as an obese–overweight marker in female samples.

Other comparisons were performed for male- and female-only samples (Appendix A). When comparing smoking status, in male samples, only one taxonomic group (Lachnospiraceae) was found as a marker for non-smokers and Acidobacteriaceae was found for men who smoked; on the other hand, in women, six groups—four of them found as makers in the general population—were found in non-smokers (Pseudoalteromonas, Pseudoalteromonadaceae, Acidobacteria, Acidobacteriia, Acidobacteriales, and Fibrobacterales–Acidobacteria group) and two markers (Microbacteriaceae, *Microbacterium*) were found in smokers. In terms of physical activity level, men and women also showed different markers: samples from men showed six taxonomic groups derived from Firmicutes as markers of active/vigorous activity, whereas women showed a set of eight markers associated with Proteobacteria and related (with the exception of *Bacillus*) as markers of sedentary activity. Finally, in the case of alcohol consumption status, the same observation was noticed in men samples; one group was found as a marker for alcohol consumption (Sphingomonales), and in women, three groups (*Citrobacter*, Enterobacteriaceae, and Enterobacteriales) were found as markers for alcohol consumption. This case is particularly interesting since, when no sex differentiation was considered, only *Citrobacter* was found (see above).

All of these results suggest that LefSe analysis could show differentiative features even if those groups have no well-established clustering patterns. These results also suggest that in some cases, groups separated by sex can contain distinctive markers that cannot be seen in the general population.

### 3.5. Different Prevalent Taxa across Different Comparisons

Previous comparisons suggest the presence of marker taxonomic groups among different lifestyles in the cohort dataset. The presence of different taxons primarily detected in most samples may also be useful to distinguish different conditions. In order to evaluate which taxonomic groups could distinguish those lifestyles, we present Venn diagrams representing common and distinctive groups among lifestyles (Appendix A), only considering “top prevalent groups” (i.e., all taxonomic groups that are present in at least 75% of the samples of each category or lifestyle).

The main highlights of the comparison within the general population (men and women) showed that, in some cases, few groups were distinct between two different conditions (Appendix A). For example, one and seventeen groups were distinct in women and men, respectively. In the case of smoking status, this was more evident, with 62 distinctive groups present in smokers but none in non-smokers, and 65 groups could differentiate between people with sedentary physical activity but none in active-vigorous physical activity. A set of 23 groups were distinctively present in people who reported not drinking alcohol but none in alcohol drinkers. Finally, seventeen groups were found in normal-low weight subjects in comparison with only one in the overweight-obese group.

This approach also found differences when men-only or women-only groups were used (Appendix A). The main highlights of the comparison within the general population (men and women) showed that, in some cases, few groups were distinct between two different conditions. In the case of smoking status, the comparison showed 17 and 29 distinct groups for non-smokers in men and women, respectively. In men and women, 26 and 27 groups, respectively, could differentiate between people with no alcohol consumption from those who consumed alcohol. In the case of physical activity, the comparison showed 37 and 47 distinct groups for active-vigorous activity in men and women, respectively. Finally, in men and women, nine and forty-two groups, respectively, could differentiate overweight-obese people from those who were not. In the case of women, nine groups were also distinctive for normal weight status (and zero groups in men). The existence of different groups between men and women across different lifestyle comparisons also suggests the potential presence of markers associating nostril microbiota composition with different lifestyles, in a sex-dependent manner.

### 3.6. Associations of Alpha Diversity with Lifestyle Attributes at the Individual Level

Multivariate models suggested that alpha diversity varied in the function of physical activity, nutritional status, smoking, and the interaction between nutritional status and smoking, although the significant impact of those variables varied between women and men. While physical activity and nutritional status were significantly associated with all indexes of alpha diversity among women, the diversity of microbiota among men was associated with smoking and the interaction between nutritional status and smoking (Table 2). For instance, women who reported sedentary and vigorous levels of physical activity had 0.25 (*p* = 0.006; 95% CI = [−0.43, −0.08]) and 0.34 (*p* = 0.01; 95% CI = [−0.60, 0.09]) units decreased on the predicted Shannon index, respectively, compared with women with active physical activity (Table 2). Similarly, a decrease of 4.77 (*p* = 0.004; 95% CI = [−7.94, −1.60]) and 6.03 (*p* = 0.008; 95% CI = [−10.43, −1.64]) of the predicted equivalent number of taxonomic groups was observed among women who reported sedentary and vigorous physical activity, respectively, compared to women who reported active physical activity (Table 2). The Simpson’s index and the inverse of the Simpson’s index showed similar results for physical activity among women. Nutritional status was also only significantly associated with alpha diversity among women, where women with obesity, compared with those of normal weight, showed a 0.18 unit increase in the predicted Shannon’s index (*p* = 0.01, 95% CI = [0.03, 0.32]), 3.0 higher equivalent numbers of taxonomic groups (*p* = 0.01, 95% CI = [0.65, 5.36]), a 0.01 unit decrease in Simpson’s index (*p* = 0.04, 95% CI = [−0.02, <0.00]), and a 1.31 unit increase in the inverse Simpson’s index (*p* = 0.05, 95% CI = [>0.00, 2.6]) (Table 2).

Smoker men were associated with a lower Shannon’s index (OLS coefficient = −0.38, *p* = 0.007, 95% CI = [−0.64, −0.11]), lower numbers of equivalent taxonomic groups (OLS coefficient = −6.36, *p* = 0.003, 95% CI = [−10.42, −2.30]), a higher Simpson’s index (OLS coefficient = 0.02, *p* = 0.04, 95% CI = [>0.00, 0.04]), and a lower inverse Simpson’s index (OLS coefficient = −3.46, *p* = 0.01, 95% CI = [−6.08, −0.83]) compared with non-smoker men (Table 2). The models also suggested that the associations between alpha diversity and lifestyle attributes depend on nutritional status. The diversity of the microbiota of people with normal weight is affected by physical activity, smoking, and the consumption of alcohol, while the increase in the diversity of microbiota is only associated with the consumption of alcohol among overweight and obese people (Appendix A). The results suggest that sedentary and vigorous physical activity decreased the diversity compared with active physical activity among people of normal weight (Appendix A). Smokers with normal weight had lower diversity of microbiota than non-smokers with the same nutritional status (Appendix A). A similar association was found with the consumption of alcohol (Appendix A). Other interactions between lifestyle attributes were also tested, but they were not statistically significant. The results of the multivariate regressions for all samples are offered in Appendix A.

## 4. Discussion

The microbiota in the human body has unique characteristics; their influence on our health has been gradually revealed. Despite the upper respiratory tract microbiota being the primary source of the lung microbiome, studies of the microbiota present in the airways have been dominated by works focused on the lower airways. On the contrary, the nasal microbiota has been scarcely explored [17,35].

This study revealed that the alpha diversity of nasal microbiota is associated with lifestyle attributes, but these associations depend on sex and nutritional status. Thus, by separating our initial data into groups of males and females, we observed that the nutritional status and the amount of physical activity in females changed the alpha diversity of the nasal microbiota. In contrast with males, we observed that only smokers or males who were overweight with or without smoking had a change in the diversity of the nasal microbiota, not according to other lifestyle factors such as physical activity, nutritional status, or alcohol consumption.

As a very external part of the upper respiratory tract, with extensive contact with the skin, the nares contain a microbiome mainly composed of members of Firmicutes (currently Bacillota), Actinobacteria, and Proteobacteria (currently Pseudomonadota) [36,37]. The main genera found in the nares communities have been previously described [36,37] and include *Staphylococcus*, *Corynebacterium*, *Propionibacterium*, and *Streptococcus*, as well as *Cutibacterium*, *Lawsonella*, *Anaerococcus*, *Moraxella*, and *Dolosigranulum* [38]. Our data showed similar general behavior, despite these samples also showing the presence of *Anaerobacillus*, *Ochrobactrum*, and *Delftia*. The taxonomic genera found in half of the samples analyzed in this study were *Staphylococcus*, *Anaerobacillus*, *Corynebacterium*, *Propionibacterium*, *Delftia*, and *Ochrobactrum*. *Staphylococcus*, *Corynebacterium*, and *Propionibacterium* are commonly found in the human nasal microbiota, and they are all part of the phylum Actinobacteria or Firmicutes [39,40]. *Anaerobacillus*, on the other hand, is a genus of anaerobic bacteria, which seems to be depleted, like other anaerobic bacteria, in patients infected by SARS-CoV-2, suggesting that the nasopharyngeal microbiota, as in any respiratory infection, plays a role in the clinical course of the disease [41]. On the other hand, a reduced relative abundance of *Anaerobacillus* has been described in bronchoalveolar lavages of patients with lung tumors; however, its role in the microbiota and its possible impact on health or disease is not clear [42]. *Delftia* and *Ochrobactrum* are both members of the phylum Proteobacteria, and they have been identified in the human nasal microbiota, but they are less abundant compared to Actinobacteria and Firmicutes [43,44]. On the other hand, according to LDA analysis, the Acidobacteriales order was highly represented in the smoking volunteers. However, to the best of our knowledge, this family has not been reported in nasal samples. Acidobacteriales are associated with soil samples [45,46], and only one case has been described in association with humans in a feces sample [47].

Within the results obtained when making comparisons only using the top prevalent groups, we were struck by the presence of unique taxonomic groups in some of these comparisons. When we compared the taxonomic groups between men and women, we could see that, in women, the Phyllobacteriaceae group was exclusive. Although there are no previous reports indicating a role for *Phyllobacterium* in disease, Wen et al. observed a higher abundance of *Phyllobacterium* spp. in nasopharyngeal samples from children infected with influenza A virus [13], and similarly, an increase in the *Phyllobacterium* genus was detected in the throat microbiota of children with cystic fibrosis, in whom a dysfunction of their immune system in the airway has been reported [48].

Regarding the association of *Phyllobacterium* in women, very few studies have addressed its detection in women. For example, Li et al. described the vaginal microbiota in healthy women during different gestation periods, observing that Lactobacillus is the dominant bacterium, and the species composition is relatively constant during normal pregnancy, being replaced after pregnancy by different bacteria, including *Phyllobacterium*, possibly associated with fluctuations in estrogen levels, birth trauma, and vaginal surgery, increasing the possibility of infection [49,50]. On the other hand, Onywera et al. described the cervical microbiota of South African women of reproductive age with and without a high risk of human papillomavirus (HPV) infection, where the *Phyllobacterium* genus was significantly abundant in women without a high risk of HPV infection [51]; our study is the first work that associates the presence of this atypical pathogen with the nasal microbiota of women.

Another interesting result obtained in this type of comparison was when comparing only women who did and did not participate in physical activity. Observing Burkholderiaceae as the only taxonomic group in the condition of sedentarism, the results were similar to those observed in the LDA. The relationship between this family of bacteria and the nasal microbiota is discussed below.

In our study, we found differences between males and females according to their lifestyles. Sex effects have been described to exist in the association of airway microbial markers and asthma [52]. Differences in the microbiota composition between males and females have been widely described in the gut [53,54,55,56,57,58], mainly due to the differences observed between the sexes in the human intestine [23]. Sex differences in the gut microbiome are partially driven by sex hormones, contributing to sex differences in immunity and susceptibility to infections and chronic diseases [57,59,60,61,62,63]. The interaction between microbiota, sex hormones, and the immune system is denominated as microgenderome, and it involves bidirectional interactions between the microbiota, hormones, immunity, and disease susceptibility [57,60,61].

A loss of microbiota diversity has been associated with several diseases [64,65], while increased diversity is associated with a better state of health [66]. An increase in microbiota diversity has been reported in professional sportspeople [67] and also in animal models in which physical activity was vigorous [68,69]. Contrary to this, our findings indicate that vigorous activity, as well as sedentary activity, produced a decrease in microbiota diversity in females and an increase in microbiota diversity in those with vigorous levels of activity. Our findings might be an example of the intermediate disturbance hypothesis [70] in upper respiratory tract microbiota. The intermediate disturbance hypothesis predicts that the diversity of taxonomic groups is expected to be highest at intermediate levels of disturbances, explained by the active levels of physical activity influencing the competition–colonization trade-off, wherein less competitive taxonomic groups are able to reproduce and colonize the respiratory tract before being eliminated by more competitive taxonomic groups, resulting in a coexistence of the two with the higher diversity of taxonomic groups. Complementing this information when carrying out the LDA score, four taxonomic groups as markers associated with sedentary activity groups were only found for females; these groups were *Bacillus*, *Sphingomonas*, Burkholderiaceae, and *Ralstonia*. All of these genera of bacteria have previously been found to be present in the human nasal microbiota, although their prevalence and abundance may vary between individuals [71,72]. Some species of *Sphingomonas* are considered possible causes of nosocomial infections [73]. Burkholderiaceae is a family that includes several genera, including *Burkholderia*, *Paraburkholderia*, and *Caballeronia*. *Burkholderia sp*. is known for its ability to degrade a wide range of organic compounds [74], and it can play a role in the breakdown of mucins as a carbon source for growth [75]. Some *Burkholderia* species are considered opportunistic pathogens, such as the case of the *Burkholderia cepacia* complex (Bcc), which is a group of genetically distinct but phenotypically similar bacteria divided into at least nine species, which, in patients with cystic fibrosis, induce the local release of pro-inflammatory factors and immunopathological disorders [76]. Moreover, *Ralstonia* can be found in a wide range of environments, including soil, water, and the human respiratory tract. Although some species of *Ralstonia* can cause infections in humans or be associated with chronic diseases [77], the exact role of these bacteria in the nasal microbiota and their impacts on human health is not yet fully understood. In the case of the genera *Ralstonia*, *Sphingomona*, *Burkholderia*, and *Rhizobium*, it cannot be ruled out that their detection stems from the possibility of contamination by manipulation in the DNA extraction or by the use of contaminated DNA extraction kits, as previously reported by Salter et al. [78] in samples of low-biomass microbiota. Unfortunately, due to this type of analysis, no samples of individuals can be considered negative controls. However, the possibility of contamination is reduced thanks to the random order processing of the samples [78].

On the other hand, multivariate models have also suggested that the diversity of microbiota in men is associated with smoking and the interaction between being overweight and smoking. Our results support that the nose microbiota of adults has been proven to be altered in those with obesity [79]. Obesity induces changes in the composition of the microbiota, whereas an additive effect is observed in obese asthma patients. The microbiome has become a fundamental topic to include in the assessment of nutritional status and, therefore, in the study of human health. Therefore, the study of healthy adults has been promoted as a reference population. In the LDA score, five taxonomic groups (*Rothia*, Micrococcineae, *Moraxella*, Moraxellaceae, and Pseudomonadales) were only found as markers associated with the obese-overweight nutritional status in males. These taxonomic groups are all genera or orders of bacteria that are commonly found in the nasal microbiota of both men and women [80]. There is little evidence of any specific effect of these taxonomic groups on the male nasal microbiota. Most studies have considered groups of people without segregating by sex. The *Rothia* genus, which belongs to the Microccineae family, has been shown to produce antimicrobial compounds that can inhibit the growth of pathogenic bacteria, such as *M. catarrhalis*, and it is therefore thought to contribute to protection against respiratory infections [81,82]. Increasing evidence regarding the role of *Moraxella* in the respiratory tract microbiota has accumulated in recent times. *Moraxella* and Moraxellaceae have been associated with respiratory infections, including acute otitis media in children [81,83], and even its detection in infants with wheezing has been strongly associated with the development of persistent asthma in adulthood [81,84]. However, *Moraxella* can also be associated with a protective role as part of the nasal microbiota. Yu et al. observed that *Moraxella* occupied the largest proportion of healthy children, and the authors suggested that *Moraxella* may be associated with better outcomes after COVID-19 infection by modulating inflammation through the regulation of amino acid metabolism pathways [85]. Additionally, some species of *Pseudomonas*, such as *Pseudomonas aeruginosa*, have been described as an opportunistic pathogen associated with respiratory infections, particularly in immunocompromised individuals [86]; a high abundance of *Pseudomonas* in the nasal microbiome may predispose the host to severe respiratory viral infection [87]. Recently, Rhoades et al. observed that, in nasal swabs from SARS-CoV-2 individuals, the abundance of *Pseudomonas aeruginosa* and other pathobionts increases with SARS-CoV-2 viral RNA load, which could contribute to the increased incidence of secondary bacterial infection [87].

In conclusion, the composition and diversity of the microbiota of the upper respiratory tract are associated with lifestyles, but the associations depend on sex and nutritional status. Future studies of the airway microbiome may provide a better resolution if data are stratified for differences in sex and nutritional status.

## Figures and Tables

**Figure 1 microorganisms-11-01635-f001:**
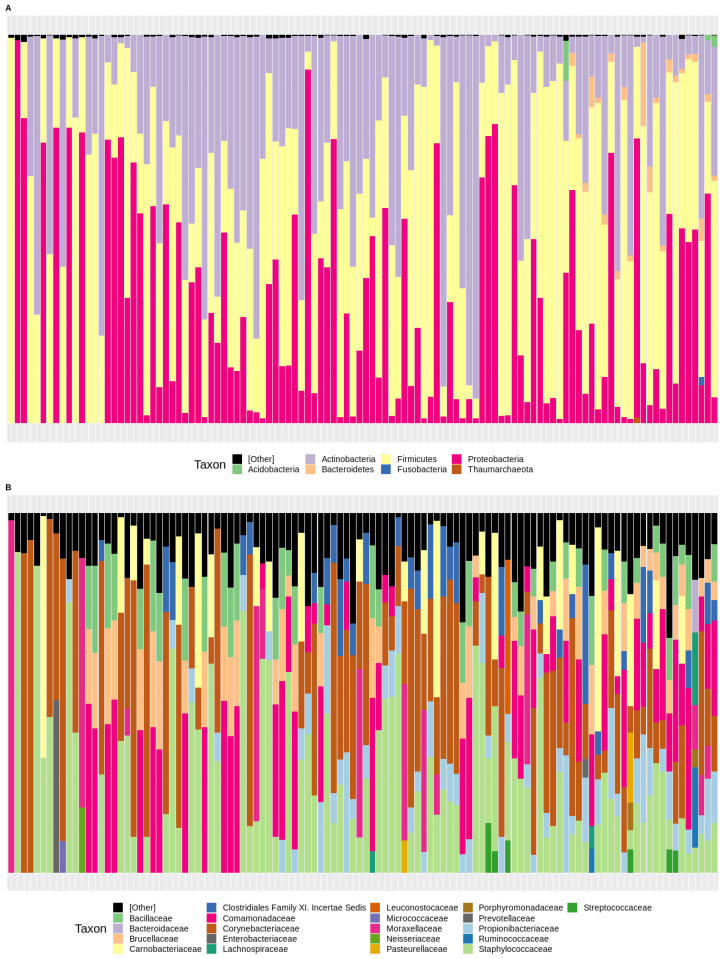
Barplots taxonomic groups representing the most abundant phyla (**A**) and the most abundant families (**B**) among the samples analyzed from the healthy adults in Santiago, Chile. Most abundant phyla were considered from a minimum of 1% in at least one sample, whereas most abundant families were considered those groups with a minimum abundance of 5% in at least one sample.

**Table 1 microorganisms-11-01635-t001:** Most prevalent genera among nose samples. Prevalence was estimated as the percentage of samples in which each taxonomic group was detected.

Genus	Phylum	Median	Standard Deviation	Prevalence (%)
*Staphylococcus*	*Firmicutes*	17.23	20.95	89.09
*Corynebacterium*	*Actinobacteria*	18.11	23.33	82.73
*Propionibacterium*	*Actinobacteria*	6.38	6.77	79.09
*Delftia*	*Proteobacteria*	17.67	15.43	64.55
*Ochrobactrum*	*Proteobacteria*	9.05	7.90	56.36
*Anaerobacillus*	*Firmicutes*	8.49	6.28	54.55
*Anaerococcus*	*Firmicutes*	3.30	3.48	49.09
*Peptoniphilus*	*Firmicutes*	2.72	2.69	36.36
*Dolosigranulum*	*Firmicutes*	14.96	16.38	34.55
*Streptococcus*	*Firmicutes*	2.11	2.93	25.45
*Bacillus*	*Firmicutes*	2.04	1.38	25.45
*Moraxella*	*Proteobacteria*	8.64	26.60	20.91
*Finegoldia*	*Firmicutes*	2.70	1.89	19.09
*Mesorhizobium*	*Proteobacteria*	1.69	0.84	13.64
*Citrobacter*	*Proteobacteria*	1.51	0.50	12.73
*Sphingomonas*	*Proteobacteria*	1.40	1.15	12.73
*Rhizobium*	*Proteobacteria*	1.40	0.88	11.82

**Table 2 microorganisms-11-01635-t002:** Association between alpha diversity indexes of microbiota and lifestyle attributes of women (n = 63) and men (n = 47).

Variables	Shannon’s		Equivalent Number		Simpson		Inverse of Simpson	
Lifestyle Attributes	Women	Men	Women	Men	Women	Men	Women	Men
Physical activity								
Sedentary vs. active	−0.25 **	−0.01	−4.77 **	−0.39	0.02 **	−0.00	−3.09 **	0.25
	(−0.43, −0.08)	(−0.35, 0.33)	(−7.94, −1.60)	(−5.92, 5.13)	(0.01, 0.03)	(−0.03, 0.02)	(−4.77, −1.41)	(−3.18, 3.68)
Vigorous vs. active	−0.34 **	0.07	−6.03 **	0.94	0.02 *	−0.01	−3.68 **	1.07
	(−0.60, −0.09)	(−0.26, 0.39)	(−10.43, −1.64)	(−4.42, 6.30)	(0.01, 0.04)	(−0.03, 0.02)	(−6.10, −1.26)	(−2.31, 4.46)
Nutritional status								
Overweight vs. normal weight	−0.03	−0.10	−0.56	−1.80	0.00	0.01	−0.39	−1.03
	(−0.20, 0.15)	(−0.32, 0.12)	(−3.42, 2.31)	(−5.57, 1.97)	(−0.01, 0.01)	(−0.01, 0.02)	(−1.99, 1.22)	(−3.25, 1.19)
Obese vs. normal weight	0.18 *	0.06	3.00 *	1.00	−0.01 *	−0.00	1.31 *	0.31
	(0.03, 0.32)	(−0.22, −0.35)	(0.65, 5.36)	(−3.94, 5.94)	(−0.02, <0.00)	(−0.02, 0.01)	(>0.00, 2.62)	(−2.48, 3.10)
Smoking	−0.11	−0.38 **	−2.10	−6.36 **	0.00	0.02 *	−0.80	−3.46 *
	(−0.31, 0.10)	(−0.64, −0.11)	(−5.57, 1.36)	(−10.42, −2.30)	(−0.01, 0.02)	(0.00, 0.04)	(−2.86, 1.25)	(−6.08, −0.83)
Interaction between nutritional status and smoking								
Overweight and smoking	0.01	0.57 *	−0.65	10.04 *	−0.00	−0.03 *	−0.58	5.35 *
	(−0.33, 0.34)	(0.12, 1.02)	(−6.69, 5.40)	(1.40, 18.68)	(−0.02, 0.02)	(−0.06, −0.01)	(−3.98, 2.82)	(1.14, 9.56)
Obese and smoking	^	0.23	^	4.10	^	−0.01	^	1.44
		(−0.31, 0.77)		(−5.27, 13.47)		(−0.05, 0.03)		(−3.57, 6.44)
R-squared	0.15	0.20	0.19	0.21	0.13	0.15	0.21	0.18

Note: Cells show ordinary least-square coefficients and, in parenthesis, the confidence interval at 95%. Coefficients report the change in the index of alpha diversity for a unit change in the lifestyle attribute. Statistically insignificant variables were omitted from the table, such as the consumption of alcohol, medication use, age, and square of age. ^ refers to variables omitted in the analysis. ** and * refer to *p* < 0.01 and *p* < 0.05, respectively.

## Data Availability

File with taxonomic composition for all samples have been deposited in the Zenodo repository under the accession number 10.5281/zenodo.8035071 with the sample ID from 103204373_NA0013000242_1_nose to 993204850_NA0013000214_1_nose, link: https://zenodo.org/record/8035071, accessed on 12 June 2023.

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
