# Peer review of "Microbiota Profile of the Nasal Cavity According to Lifestyles in Healthy Adults in Santiago, Chile"

_microorganisms, 2023, doi:10.3390/microorganisms11071635_

Round 1
Reviewer 1 Report
The work presented by Toro-Ascuy et al. is very interesting and provides novel information on the microbiome composition of healthy adults in a previously unexplored population. The introduction is clear and informative.
The authors must address several methodological issues before the article can be accepted for publication. The overall analysis flow, ideas, and methods used by the authors are adequate. Still, their application is flawed and needs to be reviewed carefully because it may have an effect on the interpretation of the results.
My main concern with the study is the lack of reproducible methods. There are no sequences reported or even tables with the taxonomic composition used by the authors for the analysis (as supplementary files, for example). The authors cite a previously published paper for the methods but do not provide anything that allows the reproducibility of their analysis. Even more, some of the methods used in that publication are also unavailable, so I recommend starting from the raw data and reprocessing to allow reproducibility and a more curated analysis.
For example, common steps used in the analysis of 16S data, such as the removal of possible contaminants (which is not possible in this study due to the lack of negative controls), or at least the removal of low prevalence taxa that could affect the outcomes of the statistical analysis.
The analysis of the microbiome composition also requires extensive revision. In particular, the LEfSe analysis needs to be checked and performed again. Looking at the Supplementary Material, some results do not make sense, for example, in the comparison between Sex, where in Female, the markers are root (all the taxa?), cellular organisms (all the taxa?), and Bacteria. This suggests that the analysis was not performed adequately using the proper taxonomic levels in the LEfSe dataset. Because a large part of the study relies on the comparison performed using LEfSe, this must be revisited before the paper can be accepted for publication. Also, this same issue probably affects the PCA analysis because the complete matrix was used for PCA, not considering the different taxonomic levels.
The other important concern the authors must address is the possible contaminants. Due to the experimental design, no negative controls were included that allowed the removal of likely contaminant taxa from the sampling process and kits. Because nasal samples are complex to process and have less material than fecal samples, they are more prone to contamination. Some results suggest that the differences in groups are due to taxa not necessarily present in the nasal cavity. Examples of these taxa are Rhizobium and Ralstonia, among others. I recommend the authors address this issue and discuss the possibility that these are not residents of the nasal cavity. Also, I recommend checking the current literature such as Salter et al. (https://bmcbiol.biomedcentral.com/articles/10.1186/s12915-014-0087-z)
Some additional recommendations to the authors:
- More reproducible methods (as mentioned before) with more information. For example, how the alpha diversity was calculated (software or package used) is not clear.
- Incorporate the variance of each component on the PCA plots.
- I suggest the authors improve font and point sizes for the bar plots (and visualizations in general). Also, to consider the readers' possible color deficiencies at the moment of selecting the colors in the plots.
Reviewer 2 Report
This is a comprehensive article exploring the association between lifestyles and microbiota profiles in the upper respiratory tract of healthy adults in Chile.
Some of my specific comments are below.
1.Line 122,can the authors justify the usage of default options (e.g., LDA)?
2.Line 134, why only interactions between nutritional status and smoking are considered but no other possible interactions.
3.line 335, the “-“ between those negative numbers are misleading.
Line 351, “no statistically significant variables were omitted. “ is misleading, suggest the author change to “statistically insignificant variables were omitted.”
Round 2
Reviewer 1 Report
Thank you to the authors for submitting and improved version of the manuscript. Most of the issues were resolved, and I only have a few comments to improve in the final version:
- Check the format of the references; some (in particular, the new ones) have the wrong format.
- Missing information on the methods. What version of the SILVA database was used? Also, include the proper references to the used R packages.
- Consistency on name translation. In section 2.1, the authors referred to the University in English (Autonomous University), while in section 2.6, they used the correct name.
- Table 1, How is the prevalence defined on this dataset?
- Line 302, Citrobacter should be in italics.
- Table 2, What are the values shown? This should be explained in the table legend.
- Table S1, What are the values shown? This should be explained in the table legend.
